# p53 and FBXW7: Sometimes Two Guardians Are Worse than One

**DOI:** 10.3390/cancers12040985

**Published:** 2020-04-16

**Authors:** María Galindo-Moreno, Servando Giráldez, M. Cristina Limón-Mortés, Alejandro Belmonte-Fernández, Carmen Sáez, Miguel Á. Japón, Maria Tortolero, Francisco Romero

**Affiliations:** 1Departamento de Microbiología, Facultad de Biología, Universidad de Sevilla, E-41012 Sevilla, Spain; mgalindo1@us.es (M.G.-M.); sgiraldez@salk.edu (S.G.); cris_limon@us.es (M.C.L.-M.); abelmonte1@us.es (A.B.-F.); torto@us.es (M.T.); 2Instituto de Biomedicina de Sevilla (IBiS), Hospital Universitario Virgen del Rocío/CSIC/Universidad de Sevilla, E-41013 Sevilla, Spain; csaez1@us.es (C.S.); mjapon@cica.es (M.Á.J.); 3Departamento de Anatomía Patológica, Hospital Universitario Virgen del Rocío, E-41013 Sevilla, Spain

**Keywords:** FBXW7, p53, tumor suppressor, cancer, proliferation, ubiquitylation

## Abstract

Too much of a good thing can become a bad thing. An example is FBXW7, a well-known tumor suppressor that may also contribute to tumorigenesis. Here, we reflect on the results of three laboratories describing the role of FBXW7 in the degradation of p53 and the possible implications of this finding in tumor cell development. We also speculate about the function of FBXW7 as a key player in the cell fate after DNA damage and how this could be exploited in the treatment of cancer disease.

FBXW7 (F-box and WD repeat domain-containing 7) is the subunit of the SCF (SKP1-CUL1-F-box protein) (FBXW7) ubiquitin ligase responsible for recruiting substrates. Polyubiquitylated proteins are then degraded by the proteasome or are involved in processes such as DNA repair, selective autophagy or signal transduction. FBXW7 is considered a critical tumor suppressor of human cancers because it degrades important oncoproteins such as c-MYC, cyclin E, MCL1, mTOR, c-JUN, PLK1, NOTCH or AURKA [1,2,3,4,5,6,7,8]. Furthermore, about 6% of analyzed human cancers carry inactivating mutations in *FBXW7* [9]. In addition, several FBXW7 mutants have also been identified in adult T-cell leukemia patients [10]. However, hitherto, the potential oncogenic activity of wild type FBXW7 has remained elusive.

In the last few months, three articles have been published by three independent laboratories, including ours, showing similar results about the regulation of p53 stability by FBXW7 [11,12,13]. Using different approaches, we found that endogenous FBXW7 interacts with and targets p53 for polyubiquitylation and proteasomal degradation. SCF(FBXW7) polyubiquitylates p53 at Lys-132 via Lys48 linkage after p53 phosphorylation on its unique FBXW7 binding motif (CDC4 phosphodegron, CPD), for which the sequence is ^31^VLSPLPS^37^ [12]. The proteasome machinery then degrades the polyubiquitylated p53. Besides the discovery of another p53 ubiquitin ligase, as a significant number of them have already been described [14], our studies show that the SCF(FBXW7)-mediated p53 degradation could have important effects on tumorigenesis. This degradation is carried out both in unstressed cells but primarily after DNA damage. Under normal culture conditions, the p53 CPD is phosphorylated by GSK3β (Ser33) [11,12] and DNA-PK (Ser37) [12]. However, after DNA damage, ATM (after ionizing radiation or etoposide treatment) [13] or GSK3β (after UV irradiation) [11] are responsible for p53 phosphorylation. The discrepancies between the detected kinases involved in p53 phosphorylation after DNA damage could underlie the different types of genotoxic agent and/or the different timings (immediately after DNA damage or later) used by each laboratory in their research.

The decrease in p53 levels after DNA damage-induced arrest promotes cell proliferation recovery. This finding could have relevant consequences for the success of the treatment of cancer with DNA-damaging agents: if the cells of a specific tumor manage to degrade p53 after genotoxic treatment they could survive and become resistant. In this regard, the analysis of breast cancer patient datasets suggested that an increase in FBXW7 reduces the survival of patients with wild type p53. However, this effect was not observed in patients carrying a non-functional p53 (mutated or lost). These data indicate that the deleterious effects of FBXW7 amplification are dependent on the degradation of a functional p53 [11]. Supporting this hypothesis, experiments carried out by Cui et al. demonstrated that inactive FBXW7 sensitizes cancer cells to radiation or etoposide treatment because it stabilizes p53 and leads to cell cycle arrest and apoptosis [13]. Moreover, p53 mutant cell lines, “untargetable” by FBXW7, were treated with doxorubicin to test their drug resistance. Long-term clonogenic, invasion and soft agar assays showed a decrease in their ability to form clones and to invade [12]. All together, these data suggest that after DNA damage, FBXW7-mediated degradation of p53 allows cell proliferation recovery and could have a potential negative effect in the outcome of treatment of cancer disease.

Conceptually, the results described above contribute to pointing out the thin line demarcating the concepts tumor suppressor and oncoprotein, since a protein may act as a tumor suppressor or as an oncoprotein depending on the cellular context. In this case, the action of FBXW7 (generally considered as a tumor suppressor) on another tumor suppressor, p53, could favor tumorigenesis. Perhaps this effect depends on the FBXW7/p53 ratio. The imbalance of this ratio would allow cell proliferation when high and prevent it when low. This could be exploited to design new therapies for cancer. For example, the use of FBXW7 inhibitors together with genotoxic agents could strongly decrease this ratio, reducing FBXW7 activity and increasing p53 levels, and, consequently, inhibiting cell proliferation. Therefore, the protein landscape of a specific cell or tissue will be key for a protein to act as an inducer or repressor of cell proliferation.

Several lines of evidence have shown that activated p53 can directly bind and activate *FBXW7* gene expression [15,16]. Hence, it is plausible that a negative feedback loop relationship exists between p53 and FBXW7. In the case of p53 accumulation after stress, it would induce *FBXW7* expression, which in turn would reduce p53 protein to basal levels, allowing the restoring of a normal cell cycle. This feedback-based regulation has been described previously in some of the reported ubiquitin ligases that target p53: MDM2, Pirh2 and COP1 [17,18,19].

Additionally, we obtained data that could help highlight the FBXW7 protein function. In our hands, immediately after ultraviolet radiation treatment, the presence of FBXW7 unexpectedly contributed to p53 stabilization [11]. This could mean that, in those early stages after treatment, FBXW7 would enhance the elimination of p53 regulators, such as MDM2 or others. As expected, p53 accumulated after DNA damage in the absence of FBXW7, most likely because of a disruption of p53/MDM2 binding, but with the presence of FBXW7-enhanced p53 accumulation. Therefore, FBXW7 could be involved in the fine-tuning regulation of DNA damage response. This ubiquitin ligase would help arresting the cell cycle by degrading p53 regulators and releasing the cells from the arrest by degrading p53 after DNA repair. Furthermore, we previously showed that SCF(FBXW7) modulates the intra-S-phase DNA-damage checkpoint by regulating PLK1 stability [6]. We found that after UV irradiation in S-phase, SCF(FBXW7)/proteasome degrades PLK1, preventing the loading of proteins onto chromatin to form pre-replicative complexes and reducing cell proliferation. If we consider these results together, FBXW7 could be considered as a safeguard of the correct cellular response to DNA damage. On the one hand, in response to DNA damage in S-phase, it would intervene in PLK1 degradation, necessary for prolonging cell proliferation arrest and preventing the progress of the cell cycle with damaged DNA. On the other, it would contribute to the p53 accumulation at the beginning of the DNA damage response. Finally, once DNA has been repaired, it would help to resume the cell cycle (Figure 1).

The problem arises when the cell divides excessively, for example in tumor cells. In that case, FBXW7 could enhance cell proliferation after DNA damage, ablating p53 and promoting the survival of tumor cells, and thus inducing resistance to treatment. However, in this case, the role that FBXW7 plays in tumorigenesis would be more a consequence than a cause of tumor transformation.

Until now, the clinical data analyzed to study the relationship between FBXW7 amplification and p53 came only from breast cancer patient datasets [11]. To generalize the conclusions, the analysis should be extended to different types of cancer datasets. Based on the articles reviewed here, it has been proposed that the use of FBXW7 inhibitors together with genotoxic drugs may be a possible approach for cancer treatment, but in-depth studies should be done to prove the therapeutic efficacy of this treatment. Additionally, it has been proposed that there is an MDM2, FBXW7 and p53 functional relationship, which should be further studied to clarify the role of this axis in cell homeostasis. It is also possible that SCF(FBXW7) targets hotspot p53 mutants. Study of the degradation of these mutants could open the door to new approaches to treat cancer, carrying p53 with oncogenic gain-of-function mutations. Lastly, it will also be important to study the effect of FBXW7 inhibition on its oncogenic substrates and its consequences in tumors. These are several questions that should be answered in the future.

## Figures and Tables

**Figure 1 cancers-12-00985-f001:**
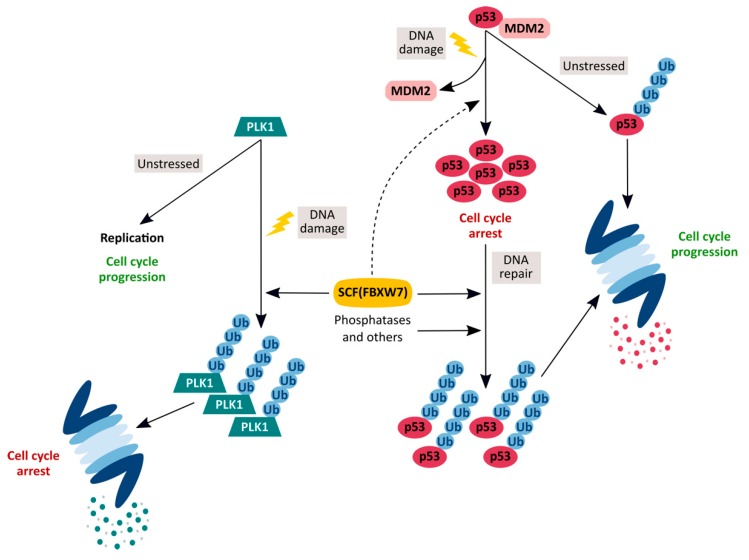
A simplified model for the role of the SCF(FBXW7) ubiquitin ligase as guardian of the correct cellular response to DNA damage. Under unstressed conditions, PLK1 phosphorylates various proteins that promote DNA replication. Upon DNA damage, SCF(FBXW7) targets PLK1 for proteasomal degradation, arresting cell cycle in S-phase. On the other hand, mainly MDM2, but also other ubiquitin ligases, maintain low levels of p53 under normal growth conditions. After DNA damage, p53 accumulates and arrests the cell cycle to allow damage to be repaired. After that, p53 levels decrease again due to SCF(FBXW7) and others, to recover cell cycle progression.

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
