# Peer review of "p53 and FBXW7: Sometimes Two Guardians Are Worse than One"

_cancers, 2020, doi:10.3390/cancers12040985_

Round 1

Reviewer 1 Report

The authors in this manuscript present a unique opinion that a tumor suppressor acting on another tumor suppressor can be harmful. They propose a mechanism for how FBXW7 mediated p53 degradation could play an important role in tumorigenesis. This is based on the recent publications from 3 different laboratories. I think that overall the idea is interesting and would be worth publication. However, I have some minor suggestions to further improve the quality of this manuscript.

  1. According to the abstract, the manuscript discusses the findings of 3 different laboratories. However, I feel that the authors comment on the other two published reports only to support their results. It would be helpful if they could point out the differences and similarities among the three reports.
  2. The mechanism of how p53 degradation is promoted by FBXW7 is not clear in this manuscript. Has it been shown by any of the three recent papers?
  3. Needs a little formatting - Line 22 has FBXW7 without parentheses but they are not consistent with this term. 
  4. I found some grammatical errors throughout the manuscript. For example, “This result, obtained analyzing several cellular models……” Since language issues are significant, I highly recommend authors to consult a native English speaker.
  5. “FBXW7 is considered a critical tumor suppressor of human cancers because it is involved in the degradation of important oncoproteins such as c-MYC, cyclin E, MCL1, 25 mTOR, c-JUN, PLK1, NOTCH or AURKA.” needs to be referenced.
  6. Line 35- It is not very clear which studies they are referring to.

Author Response

Response to Reviewer 1 Comments

  1. According to the abstract, the manuscript discusses the findings of 3 different laboratories. However, I feel that the authors comment on the other two published reports only to support their results. It would be helpful if they could point out the differences and similarities among the three reports.

We think that the three articles are very similar and, therefore, in the present manuscript we only describe the differences that complement the common idea. But, as requested, we have added other differences among the three articles.

  1. The mechanism of how p53 degradation is promoted by FBXW7 is not clear in this manuscript. Has it been shown by any of the three recent papers?

In the new version, we have added the p53 degradation mechanism, as requested.

  1. Needs a little formatting - Line 22 has FBXW7 without parentheses but they are not consistent with this term.

As recommended, parentheses have been added.

  1. I found some grammatical errors throughout the manuscript. For example, “This result, obtained analyzing several cellular models……” Since language issues are significant, I highly recommend authors to consult a native English speaker.

As requested, a native English speaker (and researcher) has read and corrected the manuscript.

  1. “FBXW7 is considered a critical tumor suppressor of human cancers because it is involved in the degradation of important oncoproteins such as c-MYC, cyclin E, MCL1, 25 mTOR, c-JUN, PLK1, NOTCH or AURKA.” needs to be referenced.

References have been added.

  1. Line 35- It is not very clear which studies they are referring to.

We refer to the three articles about FBXW7/p53. In the present version, we have clarified this point.

Reviewer 2 Report

The authors have summarized and highlighted three recent reports, including the authors' paper. All three papers have consistently reported on the observation that FBXW7 degrades p53. It is nice to highlight consistent results among three independent groups. The finding is important as it provides an answer to the question as to why p53 upregulates FBXW7. In addition, the authors extensively comment on the interpretation of the results from the three papers in the context of cancer biology. Although I do not fully agree with the authors' model, the idea is interesting and the extensive discussion provides a more comprehensive explanation, which addresses some of the logical weak points.

Major comments

A gap between the title and the content of the paper

The title proposes an interesting concept. However, the meaning of “Sometimes” is not apparent even after reading the entire manuscript. In general, multi-protection systems are more stable and more fail-proof than single protection systems. Here, the authors propose incompleteness or a system error in the FBXW7-p53 regulatory unit. It is worth that the authors specify when the FBXW7-p53 regulatory unit shows a system error, which can produce a worse outcome in cancer patients.

FBXW7/p53 ratio model

I understand the benefits of simplifying the mechanisms underlying FBXW7-p53 regulation. In lines 42-45, page 1, the authors conclude that "These results suggested that the role of FBXW7 on cell survival occurs through p53.” There is no explanation of why the patients’ outcome is only dependent on p53 rather than on other targets of FBXW7, including c-Myc, cyclin E, Mcl1, mTOR, c-Jun, Notch and AURKA. It is necessary that the authors provide a more comprehensive explanation before proposing a FBXW7/p53 simplified model.

Minor comments

  1. Line 17, page 1: ”the proper functioning of the cell".

The meaning of “the proper functioning of the cell” is not clear. It would be better to be more specific.

  1. Line 24-26, page 1: "FBXW7 is considered a critical tumor suppressor of human cancers because it is involved in the degradation of important oncoproteins such as c-MYC, cyclin E, MCL1, mTOR, c-JUN, PLK1, NOTCH or AURKA."

This is already described by Chien-Hung Yeh et al. (2018) in Molecular Cancer. It should be clarified that this is a previously reported idea.

  1. Line 26-27, page 1: "Furthermore, about 6% of analyzed human cancers have inactivating mutations in FBXW7 [1]."

This sentence is citing a review paper. It is appropriate to cite the original articles here.

  1. Line 62, page 2: " A model" --> "A simplified model"

As I mentioned before, it is worth providing a simplified model for easier understanding of a complex area of research. At the same time, it is important to provide a more comprehensive and balanced overview, which includes discussion of other proto-oncogenes before proposing a simplified model. In the current case, the model proposed may conflict with current knowledge linking the tumor suppressor function of FBXW7 to degradation of oncoproteins. To avoid this issue, it may be better to refer to the model proposed as "A simplified model" rather than "A model".

  1. Line 82, page 3: "the correct functioning of the cell"

What is "the correct functioning of the cell"? It requires the imagination of the reader. Please be more specific.

Author Response

Response to Reviewer 2 Comments

Major comments

A gap between the title and the content of the paper

The title proposes an interesting concept. However, the meaning of “Sometimes” is not apparent even after reading the entire manuscript. In general, multi-protection systems are more stable and more fail-proof than single protection systems. Here, the authors propose incompleteness or a system error in the FBXW7-p53 regulatory unit. It is worth that the authors specify when the FBXW7-p53 regulatory unit shows a system error, which can produce a worse outcome in cancer patients.

Hypothetically, an increase in the amount or activity of a tumor suppressor protein is expected to produce a better outcome in cancer patients. However, breast cancer patients with wild type p53 and amplified FBXW7 showed a poor prognosis compared with those ones without amplification of FBXW7. We have clarified this point in the manuscript. So, as long as the activity of the “guardian” of the genome, p53, is not compromised, the increase of another “guardian”, FBXW7, could result in an advantage for tumor cell proliferation. We consider this is an example of the reasons why “sometimes” two guardians are worse than one.

FBXW7/p53 ratio model

I understand the benefits of simplifying the mechanisms underlying FBXW7-p53 regulation. In lines 42-45, page 1, the authors conclude that "These results suggested that the role of FBXW7 on cell survival occurs through p53.” There is no explanation of why the patients’ outcome is only dependent on p53 rather than on other targets of FBXW7, including c-Myc, cyclin E, Mcl1, mTOR, c-Jun, Notch and AURKA. It is necessary that the authors provide a more comprehensive explanation before proposing a FBXW7/p53 simplified model.

The increase of FBXW7 reduces the survival of breast cancer patients with wild type p53. That said, patients carrying p53 mutations and amplified FBXW7 do not reflect a poorer prognosis than patients with mutated p53 but without amplified FBXW7. This is why we suggest that the role of FBXW7 in survival of patients occurs through p53 rather than through other FBXW7 substrates.

Minor comments

  1. Line 17, page 1: ”the proper functioning of the cell".

The meaning of “the proper functioning of the cell” is not clear. It would be better to be more specific.

  1. Line 24-26, page 1: "FBXW7 is considered a critical tumor suppressor of human cancers because it is involved in the degradation of important oncoproteins such as c-MYC, cyclin E, MCL1, mTOR, c-JUN, PLK1, NOTCH or AURKA."

This is already described by Chien-Hung Yeh et al. (2018) in Molecular Cancer. It should be clarified that this is a previously reported idea.

  1. Line 26-27, page 1: "Furthermore, about 6% of analyzed human cancers have inactivating mutations in FBXW7 [1]."

This sentence is citing a review paper. It is appropriate to cite the original articles here.

  1. Line 62, page 2: " A model" --> "A simplified model"

As I mentioned before, it is worth providing a simplified model for easier understanding of a complex area of research. At the same time, it is important to provide a more comprehensive and balanced overview, which includes discussion of other proto-oncogenes before proposing a simplified model. In the current case, the model proposed may conflict with current knowledge linking the tumor suppressor function of FBXW7 to degradation of oncoproteins. To avoid this issue, it may be better to refer to the model proposed as "A simplified model" rather than "A model".

  1. Line 82, page 3: "the correct functioning of the cell"

What is "the correct functioning of the cell"? It requires the imagination of the reader. Please be more specific.

As requested by the referee, all the minor points have been clarified or modified.

Reviewer 3 Report

Maria Galindo-Moreno and colleagues wrote a perspective mini-review focusing on three recent papers that studied the crosstalk between the tumor suppressor p53 and the tumor suppressor FBXW7. Summarizing the overall consistent findings among those three studies, one of which coming from the Authors' lab- they are raising the provocative conclusion that FBXW7 could also act as a potential oncogene by targeting p53 for degradation at least in certain contexts and cell types. The topic is relevant and interesting. However, this version of the ‘perspective’ can be improved significantly. Below are a number of suggestions on aspects that could be included or expanded in the text (not in order of relevance).

1) Discuss the possibility of a feedback loop between FBXW7 and p53; the data on p53 controlling FBXW7 transcription is somewhat convincing, based on gene expression changes and p53 ChIP-seq data in several experiments and different human cell lines treated with different p53-inducing agents, including doxorubicin, etoposide, IR. p53-dependent regulation is apparent also in mouse cells.

2) Consider to mention the difference among FBXW7 isoforms with regards to the modulation of p53 expression.

3) Consider commenting on the possibility of functional interactions between MDM2 and FBXW7 acting on p53 protein. It has been reported that they act through ubiquitination of different p53 lysine residues, with FBXW7 targeting lysine in the DNA binding domain

4) Elaborate more on the different impact of DNA damaging treatments; for example, X-rays or doxorubicin, or etoposide treatments (are double strand breaks a common element?) seem to engage FBXW7 rapidly, while for UV apparently there is a delayed effect (and there could even be a paradoxical effect at early time points).

5) Discuss the difference in results obtained about the prominent role of Ser33 and or Ser37 phosphorylation and role of ATM versus GSK3B as upstream kinases enabling FBXW7 activity on p53.

6) Discuss more the role of FBXW7 on tuning down p53 after DNA damage, perhaps the most relevant and characteristic feature of FBW7, versus its possible roles on controlling “basal” p53 levels in untreated cells.

7) Elaborate on the possibility that FBXW7 may inhibit gain of functions exhibited by hotspot p53 mutants.

8) Broaden the discussion on the potential effect of high FBXW7 levels ablating p53 levels during DNA damage response. Is it really an unexpected “oncogenic” effect? Although this is consistent with cell proliferation and survival, what is the evidence for this in vivo? I’m alluding to the studies that p53 functions in the early phase of DNA damage response may not be its primary tumor suppressive function, at least based on mouse cancer models where p53 expression can be turned on or off. Finally, I think that among the many proposed or established FBXW7 substrates there are major oncogenes, but also other tumor suppressor genes. A broader perspective on the potential beneficial role of these tuning events for cell homeostasis could be given.

9) Consider mentioning a possible role of FBXW7 in modulating p53 pulses and periodicity of expression. The oscillation of p53 and MDM2 levels have been shown to contribute to the outcome of p53-induced response in cancer cell models.

10) A final sentence highlighting what are the key questions that should be answered next is recommended.

Also, there are a few sentences in the text that could be reworded

Line 16: “tumor formation”; not all cited paper investigated directly tumor formation

Line 17: statement on function of FBXW7 is a bit vague/generic

Line 49: why are p53 mutants unable to be targeted? Is this a consistent finding of the three papers that are reviewed?

Line 54: “close an idea”…

Line 57: “favors tumorigenesis”, again how well established is this fact in cancer models in vivo?

Figure 1: consider adding in the figure the feedback loops, p53-MDM2; p53-FBXW7

Line 88: “due to its tumor transformation”?

Author Response

Response to Reviewer 3 Comments

1) Discuss the possibility of a feedback loop between FBXW7 and p53; the data on p53 controlling FBXW7 transcription is somewhat convincing, based on gene expression changes and p53 ChIP-seq data in several experiments and different human cell lines treated with different p53-inducing agents, including doxorubicin, etoposide, IR. p53-dependent regulation is apparent also in mouse cells.

As requested, the possibility of a feedback loop between FBXW7 and p53 has been discussed in the manuscript.

2) Consider to mention the difference among FBXW7 isoforms with regards to the modulation of p53 expression.

Tripathi and colleagues’ manuscript shows the effects on p53 protein levels of the specific disruption of FBXW7 isoforms α, β or γ in HCT116 cells. Their data suggest that the effect of FBXW7 on p53 degradation is specific of the α isoform. According to the scopes addressed in the three studies analyzed in this Perspective, this is the only available information regarding the effect of each specific FBXW7 isoform on p53 expression, protein level or activity. Thus, we truly appreciate your suggestion, but we consider it would be necessary further studies to discuss that topic on the manuscript.

3) Consider commenting on the possibility of functional interactions between MDM2 and FBXW7 acting on p53 protein. It has been reported that they act through ubiquitination of different p53 lysine residues, with FBXW7 targeting lysine in the DNA binding domain

In our opinion, MDM2 and FBXW7 must be functionally related. In fact, we have already suggested it on the manuscript. But, except the information about the ubiquitylation of different p53 lysine residues by MDM2 and FBXW7, we do not have enough data to discuss it in detail.

4) Elaborate more on the different impact of DNA damaging treatments; for example, X-rays or doxorubicin, or etoposide treatments (are double strand breaks a common element?) seem to engage FBXW7 rapidly, while for UV apparently there is a delayed effect (and there could even be a paradoxical effect at early time points).

We have commented about this point in the new version of the manuscript.

5) Discuss the difference in results obtained about the prominent role of Ser33 and or Ser37 phosphorylation and role of ATM versus GSK3B as upstream kinases enabling FBXW7 activity on p53.

The different kinases involved in p53 phosphorylation and phosphorylated residues have been also discussed in the new version of the manuscript.

6) Discuss more the role of FBXW7 on tuning down p53 after DNA damage, perhaps the most relevant and characteristic feature of FBW7, versus its possible roles on controlling “basal” p53 levels in untreated cells.

The role of FBXW7 on p53 stability in untreated cells has already been mentioned in our previous paper (Galindo-Moreno et al., 2019). We have not delved into this point in the present manuscript because MDM2 is the major ubiquitin ligase causing p53 turnover in basal conditions. However, as requested, we have discussed deeper the role of FBXW7 after DNA damage.

7) Elaborate on the possibility that FBXW7 may inhibit gain of functions exhibited by hotspot p53 mutants.

This is an interesting idea and, as requested, we have commented about it in the new version of the manuscript.

8) Broaden the discussion on the potential effect of high FBXW7 levels ablating p53 levels during DNA damage response. Is it really an unexpected “oncogenic” effect? Although this is consistent with cell proliferation and survival, what is the evidence for this in vivo? I’m alluding to the studies that p53 functions in the early phase of DNA damage response may not be its primary tumor suppressive function, at least based on mouse cancer models where p53 expression can be turned on or off. Finally, I think that among the many proposed or established FBXW7 substrates there are major oncogenes, but also other tumor suppressor genes. A broader perspective on the potential beneficial role of these tuning events for cell homeostasis could be given.

Our studies establish the basis for a novel regulation of p53 by FBXW7, but in vivo experiments are still needed to further link this regulation to cancer development. The identification of the tumor suppressor protein p53 as a novel substrate of FBXW7 is just an example of the complexity to classify the functions of a protein as tumor suppressor or as oncoprotein, and we discuss this idea in the manuscript. A deeper discussion about FBXW7 controlling other tumor suppressors and its role on cell homeostasis would be interesting. Even so, we consider this question should be addressed in a more profound review. In the current “mini-review”, we prefer to focus on the new role of FBXW7 in the DNA damage response through p53 stability.

9) Consider mentioning a possible role of FBXW7 in modulating p53 pulses and periodicity of expression. The oscillation of p53 and MDM2 levels have been shown to contribute to the outcome of p53-induced response in cancer cell models.

Tripathi and colleagues lightly discussed a possible role of FBXW7 in the pulsatile p53 response during DNA damage. This is a very interesting point, but none of the three articles discussed on this Perspective studies it in detail. As consequence, we consider that it is not appropriate to include a revision in the present manuscript.

10) A final sentence highlighting what are the key questions that should be answered next is recommended.

As requested, we have added to the manuscript several additional questions that would be interesting to address in future studies.

Also, there are a few sentences in the text that could be reworded

All the minor points have been revised and those sentences reworded.

Line 16: “tumor formation”; not all cited paper investigated directly tumor formation

Line 17: statement on function of FBXW7 is a bit vague/generic

Line 49: why are p53 mutants unable to be targeted? Is this a consistent finding of the three papers that are reviewed?

These p53 mutants carry mutations in the specific residues that are needed to be targeted and ubiquitylated by SCF(FBXW7) and, therefore, to be degraded by the proteasome. This is a consistent finding in the three papers.

Line 54: “close an idea”…

Line 57: “favors tumorigenesis”, again how well established is this fact in cancer models in vivo?

Figure 1: consider adding in the figure the feedback loops, p53-MDM2; p53-FBXW7

We appreciate the recommendation, but we prefer not adding an excess of information in that model, especially if this information concerns issues that we have not studied in depth (e.g. the feedback loops). It would complicate its understanding, and our goal is to provide a simplified model for an easier comprehension.

Line 88: “due to its tumor transformation”?

Round 2

Reviewer 2 Report

The manuscript was improved with more comprehensive and balanced views. The manuscript gives us new convincing suggestions, including that FBXW7-p53 can be a vulnerability of the cell systems. I am looking forward to seeing the author’s future work.

Reviewer 3 Report

The Authors have deeply revised their perspective manuscript, including several of the suggestions from the reviewers. I believe this version of the text is somewhat clearer and provides a broader perspective, also by pointing to the next key questions that are worth being pursued.

There are a couple of minor wording issues that can be resolved in the production process:

-consider removing the word “disease” in line 19

-line 47, consider replacing “for” with “in”.

-line 66, consider rephrasing “point up the..” and add “of” after “concept”

-line 79, consider to simplify the sentence, for example “p53 accumulation after stress would induce…”